# Treatment-Interval Changes in Serum Levels of Albumin and Histidine Correlated with Treatment Interruption in Patients with Locally Advanced Head and Neck Squamous Cell Carcinoma Completing Chemoradiotherapy under Recommended Calorie and Protein Provision

**DOI:** 10.3390/cancers14133112

**Published:** 2022-06-24

**Authors:** Chao-Hung Wang, Hang Huong Ling, Min-Hui Liu, Yi-Ping Pan, Pei-Hung Chang, Yu-Ching Lin, Wen-Chi Chou, Chia-Lin Peng, Kun-Yun Yeh

**Affiliations:** 1Heart Failure Research Center, Division of Cardiology, Department of Internal Medicine, Chang Gung Memorial Hospital, Keelung 20401, Taiwan; bearty@cgmh.org.tw (C.-H.W.); min4108@cgmh.org.tw (M.-H.L.); 2College of Medicine, Chang Gung University, Taoyuan 333007, Taiwan; xianfang87@cgmh.org.tw (H.H.L.); ich2682@cgmh.org.tw (P.-H.C.); yuching1221@cgmh.org.tw (Y.-C.L.); f12986@cgmh.org.tw (W.-C.C.); 3Division of Hemato-Oncology, Department of Internal Medicine, Chang Gung Memorial Hospital, College of Medicine, Keelung 20401, Taiwan; 4Department of Nursing, Chang Gung Memorial Hospital, Keelung 20401, Taiwan; 5Department of Nutrition, Chang Gung Memorial Hospital, Keelung 20401, Taiwan; yipyng@cgmh.org.tw; 6Department of Medical Imaging and Intervention, Chang Gung Memorial Hospital, College of Medicine, Keelung 20401, Taiwan; 7Osteoporosis Prevention and Treatment Center, Chang Gung Memorial Hospital, Keelung 20401, Taiwan; 8Division of Hemato-Oncology, Department of Internal Medicine, Chang Gung Memorial Hospital, Linkou, Taoyuan 333007, Taiwan; 9Taiwan Nutraceutical Association, Taipei 104483, Taiwan; jacquelin_peng@new-health.com.tw

**Keywords:** head and neck cancer, concurrent chemoradiotherapy, treatment toxicity, metabolites, albumin, histidine, treatment interruption, nutrition, calorie, protein, body composition

## Abstract

**Simple Summary:**

Treatment interruption during concurrent chemoradiotherapy (CCRT) jeopardizes the outcomes of patients with locally advanced head and neck squamous-cell carcinoma (LAHNSCC), which is associated with increased toxicity during CCRT. Nonetheless, we found that under the provision of recommended calories (25 kcal/kg/day) and protein (1 g/kg/day) throughout the treatment course, treatment-interval changes in serum albumin and histidine levels—not treatment toxicity—greatly contributed to the interruption of treatment in patients with LAHNSCC. We believe that serum levels of albumin and histidine should be integrated into the routine nutritional assessment of patients undergoing CCRT.

**Abstract:**

We investigated risk factors for treatment interruption (TI) in patients with locally advanced head and neck squamous-cell carcinoma (LAHNSCC) following concurrent chemoradiotherapy (CCRT), under the provision of recommended calorie and protein intake; we also evaluated the associations between clinicopathological variables, calorie and protein supply, nutrition–inflammation biomarkers (NIBs), total body composition change (TBC), and a four-serum-amino-acid metabolite panel (histidine, leucine, ornithine, and phenylalanine) among these patients. Patients with LAHNSCC who completed the entire planned CCRT course and received at least 25 kcal/kg/day and 1 g of protein/kg/day during CCRT were prospectively recruited. Clinicopathological variables, anthropometric data, blood NIBs, CCRT-related factors, TBC data, and metabolite panels before and after treatment were collected; 44 patients with LAHNSCC were enrolled. Nine patients (20.4%) experienced TIs. Patients with TIs experienced greater reductions in hemoglobin, serum levels of albumin, uric acid, histidine, and appendicular skeletal mass, and suffered from more grade 3/4 toxicities than those with no TI. Neither increased daily calorie supply (≥30 kcal/kg/day) nor feeding tube placement was correlated with TI. Multivariate analysis showed that treatment-interval changes in serum albumin and histidine levels, but not treatment toxicity, were independently associated with TI. Thus, changes in serum levels of albumin and histidine over the treatment course could cause TI in patients with LAHNSCC following CCRT.

## 1. Introduction

Most patients with head and neck cancer (HNC) present with a locally advanced stage at the time of diagnosis [1]. Patients with locally advanced head and neck squamous-cell carcinoma (LAHNSCC) typically require a multidisciplinary treatment approach, including surgery, chemotherapy, and radiotherapy (RT). To improve treatment outcomes, concurrent chemoradiotherapy (CCRT) is used either as adjuvant therapy following surgery or as curative-intent primary CCRT therapy [2,3]. Nonetheless, CCRT often causes severe toxicity, resulting in treatment interruption and inferior outcomes [4,5,6,7,8,9,10,11]. Only 60% of patients complete the CCRT protocol through the administration of three cycles of 100 mg/m^2^ cisplatin concurrently with RT [12,13,14,15]; over 50% and approximately 10–20% of patients experience at least one day and one week or more of treatment interruption, respectively [10,16,17,18]. There is a reduction of 0.7–1.4% in local tumor control for one-day treatment interruption, which rises to 14–20% for one week in patients with HNC undergoing RT [19,20]. In these patients, treatment interruption causes a decrease in 5-year survival rates from 61% to 28% [21], and an additional day break lowers the 5-year relapse-free survival rate by 1–2% after an RT interruption duration of over 6 days [22]. Thus, even a short RT break or just a one-day increase in the interruption period may have significant negative consequences. Therefore, it is essential to avoid or minimize interruption events during RT and CCRT courses.

The negative impact of RT interruption on treatment outcomes is attributed to repopulation of residual tumor cells—especially cancer stem cells, which proliferate indefinitely and cause tumor recurrence when RT is delayed or interrupted [9,23]. Accumulating evidence shows that under a standard and conventional fractioning radiation course scheduled for 5 days per week over 6–8 weeks, the major causes of RT interruption events are statutory/regular holidays, machine problems such as maintenance work and unexpected breakdown, waiting time for RT initiation after surgery, and patient-related factors—including non-compliance, psychological stress, insufficient knowledge about cancer and treatment, and socioeconomic problems—among patients with HNC undergoing RT [17,18,24]. To address this issue, several strategies have been successfully implemented over the past three decades to reduce treatment interruptions [17,18,24]. Two audits were conducted in the United Kingdom in 2002 and 2008 [17,18]. In the second audit, it was found that formulating suitable policies and guidelines—including setting up machinery maintenance, provision of an alternative backup linear accelerator machine, and prudent treatment scheduling with a compensation program—shortened the waiting time for patients to start RT after surgery, and increased the percentage of patients completing treatment within 2 days of the scheduled time [18]. Additionally, Chen et al. used pretreatment nursing consultation and adequate support from social and administrative resources to enhance patients’ tolerance to adverse effects and enable compliance with the treatment process, reducing RT interruptions in patients with nasopharyngeal carcinoma [24]. Currently, these planned (e.g., statutory/public holidays, machine maintenance, the time interval between surgery completion and RT initiation) and unplanned interruptions (e.g., machine faults, patient non-compliance) have been decreased in clinical practice due to effective preemptive administrative work and therapeutic schedule arrangement.

Although these strategies greatly reduce treatment interruptions, cumulative treatment toxicity from RT and chemotherapy—such as ulcerative oral mucositis and xerostomia—is still an important issue among patients afflicted with unplanned interruptions [9,16,17,18,25]. Various effective strategies—such as radiation course modulation, chemotherapy delivery modification, and diet therapy—reportedly improve these toxicities [9]; however, these results pertaining to the effects of radiation morbidity on treatment interruption should be interpreted with caution. First, most of these studies were conducted retrospectively among heterogeneous patient populations with mixed tumor stages, varying radiotherapy regimens, and different definitions of treatment interruption. These variations cause a large difference in the incidence of treatment toxicity causing interruption, varying from 2% to 20% [17,18,19]. Second, patients’ characteristics, extent of cancer involvement, and status of nutrition and inflammation are associated with the incidence and severity of treatment toxicities [26,27,28,29,30,31,32,33,34,35]. Unfortunately, few studies have investigated the interplay between these variables, treatment toxicity, and interruption. Furthermore, adding concurrent chemotherapy to RT enhances tumor control and survival outcomes, and also increases treatment-related toxicity and treatment interruptions [9]. Moreover, some reports show that treatment delay may not always contribute to worse outcomes in patients with locally advanced head and neck cancers following modifications in the chemotherapy regimen and radiation protocol over the treatment course [36,37], suggesting that certain therapeutic modes delivered during CCRT could affect treatment interruption. Finally, intensive nutritional support is mandatory for patients with LAHNSCC undergoing CCRT [27]. The European Society for Clinical Nutrition and Metabolism (ESPEN) suggests an energy intake of at least 25–30 kcal/kg/day for each patient, and recommends feeding tube usage to maintain calorie supply over the treatment course [38]. However, the effect of the recommended energy intake and feeding tube placement on treatment interruption during CCRT has seldom been addressed.

Alterations in total body composition (TBC), assessed by dual-energy X-ray absorptiometry (DXA), are commonly observed in cancer patients in response to various physiological and pathological stimuli, such as aging, sex, intercurrent illness, metabolic disturbance, and therapy; therefore, they can appropriately reflect the status of nutritional and inflammatory changes caused by cancer or treatment. We observed a significant loss in lean body muscle (LBM) and total fat mass (TFM) in patients with LAHNSCC during CCRT [39,40]. Additionally, the pretreatment level and treatment-interval changes in TBC are associated with increased treatment-related toxicity, early treatment failure, recurrence-free survival, and two-year mortality rate among LAHNSCC patients undergoing CCRT [39,40,41,42]. Therefore, further research is needed to investigate the effect of TBC changes on treatment interruption in patients with LAHNSCC undergoing CCRT.

Metabolomics, which is a useful tool for assessing biochemical changes in tissues and bodily fluids, as well as for understanding complicated disease processes, is used for patients with HNC [43,44,45,46,47]. A comprehensive review conducted by Shin et al. confirmed different metabolite profiles between healthy subjects with no cancer and patients with HNC using various sample types and metabolomic platforms [43]. Although the results of this research improve our knowledge of the possible pathogenesis of HNC, they are not widely applicable in daily practice, probably because of the limited availability of technological platforms such as proton nuclear magnetic resonance and mass spectrometry; moreover, there are complex interpretations of multi-metabolite measurements and inherent inconsistencies of metabolite profiles from different tissue sample collections [44,45,46,47]. Recently, we utilized ultrahigh-performance liquid chromatography (UPLC) to develop a simple and easy-to-read metabolite panel by measuring serum levels of only four amino acids (i.e., histidine, leucine, ornithine, and phenylalanine (HLOP)) [48,49]. This panel represents the nutritional status, muscle synthesis and breakdown, and the amount of nitrogen waste from amino acids via the urea cycle; it has proven its prognostic value in patients with heart failure [48,49], severe infection [50], chronic pulmonary obstructive disease [51], and chronic kidney disease [52]. However, the clinical role of this HLOP-based panel in LAHNSCC patients undergoing CCRT has not yet been addressed.

Therefore, a prospective observational study is required to address these issues. We prospectively enrolled a homogeneous group of patients with LAHNSCC (stage III, IVA, or IVB). They received a standard CCRT protocol that delivered RT at a fraction daily for 5 days per week over 6–8 weeks concurrently with weekly cisplatin infusion chemotherapy. To reduce treatment non-compliance and machine faults, all participants consulted psychiatrists and social workers, and our institution prepared a backup linear accelerator machine for breakdown/maintenance. Following the ESPEN guidelines, each cancer patient was requested to have a calorie intake of at least 25 kcal/kg/day with 1.0 g/kg/day of protein during the CCRT course to avoid malnutrition. Under the provision of recommended daily calories and protein throughout the treatment course, we aimed to identify potential factors contributing to treatment interruption in patients with LAHNSCC who completed the CCRT course, by simultaneously analyzing all covariates, including clinicopathological variables, anthropometric data, blood nutrition–inflammation biomarkers (NIBs), treatment-related profiles, DXA-associated measurements, and serum HLOP metabolic profiles.

## 2. Materials and Methods

### 2.1. Patient Recruitment

We prospectively recruited eligible patients with histologically proven LAHNSCC originating in the oral cavity, oropharynx, hypopharynx, and/or larynx between January 2018 and July 2019. The head and neck cancer committee confirmed the tumor stage according to the 8th edition of the American Joint Committee on Cancer (AJCC)’s staging system and criteria for inclusion and exclusion. LAHNSCC includes stages III (T1-2, N1 or T3, N0-1), IVA (T4a, N0-1 or T1-4a, N2), and IVB (any T, N3, or T4b, any N). The inclusion criteria included Eastern Cooperative Oncology Group (ECOG) performance status score ≤2, age <75 years, sufficient hematopoietic or organ function, and p-16-negative expression in the tumor specimen. Patients who met the following criteria were excluded: (1) one of the following systemic illnesses: heart failure with New York Heart Association Classification IV, severe chronic obstructive pulmonary disease, major gastrointestinal disorders, decompensated liver cirrhosis with intractable ascites or hepatic encephalopathy, end-stage renal disease, uncontrolled diabetes mellitus, enduring infections, or autoimmune diseases; and (2) regular use of medications that could markedly interfere with metabolism or weight, such as steroids or megestrol acetate.

We obtained written informed consent from cancer patients upon enrollment. This study was approved by the Institutional Review Board of Chang Gung Memorial Hospital, Taiwan (approval numbers: 101-4047B and 201700158B0), and was conducted in compliance with the Good Clinical Practice Guidelines and the Declaration of Helsinki.

### 2.2. CCRT Treatment

Patients with LAHNSCC received either postoperative adjuvant CCRT for oral cavity cancer or curative-intent primary CCRT for unresectable non-oral-cavity cancer. Adjuvant CCRT was administered based on the following pathological features: (1) extranodal extension, (2) positive surgical margin, or (3) at least three minor risk factors, including pT4, pN1, poorly differentiated histology, invasion of blood vessels, lymphatic drainage, or perineural space and close margin ≤4 mm. Intensity-modulated RT at a dose of 60–72 Gy in 30–36 fractions—a fraction daily for 5 days per week over 6–8 weeks—with concurrent chemotherapy with weekly cisplatin (40 mg/m^2^), was administered. Patients with oral cavity cancer were required to initiate the postoperative adjuvant CCRT program within 6 weeks of the completion of surgery. Patients with non-oral-cavity cancer received primary CCRT within two weeks of diagnosis.

RT dose was defined as the total radiation dose administered during CCRT; RT duration was defined as the number of days required to complete the RT dose; cisplatin dose was defined as the cumulative dose of weekly cisplatin administration.

The severity of adverse effects, including hematological and non-hematological toxicities, was graded in accordance with the Common Terminology Criteria for Adverse Events (CTCAE, version 5.0).

### 2.3. Definition of Treatment Interruption

Patients who completed the entire planned course of RT and chemotherapy but developed an RT break of over 5 days or failed to receive scheduled weekly cisplatin during the CCRT course were considered to have faced treatment interruptions. An RT break of >5 days was considered as the criterion for treatment interruption, because it was associated with the local relapse-free survival and overall survival of patients with locally advanced head and neck cancer and nasopharyngeal carcinoma [19,53,54]. Nonetheless, interruptions due to public holidays were permitted. The “incomplete CCRT group” included (1) patients who requested drop-out from the study during the treatment course; and (2) incomplete data collection—patients who could not complete the required examinations or missed scheduled tests on the planned timetable.

### 2.4. Provision of Recommended Calorie Support during CCRT

We reviewed the dietary records of patients at weekly dietitian visits. Patients could obtain oral nutritional supplements of at least 25 kcal/kg/day, with % calories from carbohydrates:lipids of 60:40 and protein of 1.0 g/kg/day during the treatment course, based on the ESPEN recommendation guidelines [38]. Therefore, feeding tube placement was required if the body weight (BW) loss was over 5% or the daily calorie intake was less than 25 kcal/kg/day for three consecutive days during the CCRT course.

### 2.5. Clinicopathological Data

Clinicopathological data—including sex, age, BW, body height (BH), ECOG performance status, comorbid illness, tumor location, tumor stage, status of tumor size (T), regional lymph node involvement (N), histological differentiation grade, records of cigarette smoking exposure, alcohol consumption, betel nut use, tracheostomy and feeding tube, treatment protocol, and toxicity profiles—were collected. Patients who were current cigarette smokers or were previously exposed to cigarette smoke were considered smokers. Patients who consumed alcohol more than 4 times per week were considered alcohol drinkers. Patients who consumed betel nuts during the previous year were considered betel nut users. We assessed the severity of the comorbidity using the head and neck Charlson Comorbidity Index (HN-CCI) [55]. Body mass index (BMI) was calculated from the BH and BW of each participant (weight in kilograms divided by the square of the height in meters, kg/m^2^).

### 2.6. Biochemical Data and Blood NIBs

Blood samples were collected after overnight fasting within 1 week before CCRT and within 3 days after CCRT. Biochemical data and NIBs—including hemoglobin levels (Hb, g/dL), white blood cell counts (WBC, 10^3^/mm^3^), platelet counts (10^3^/mm^3^), and albumin (g/dL), prealbumin (g/dL), transferrin (g/dL), creatinine (mg/dL), alanine transaminase (ALT, U/L), total bilirubin (mg/dL), uric acid (mg/dL), fasting glucose (mg/dL), total cholesterol (mg/dL), triglyceride (mg/dL), and C-reactive protein levels (CRP, mg/dL)—were measured using an auto-analyzer (Beckman, CA, USA) at the CGMH central laboratory in Keelung, Taiwan.

The estimated glomerular filtration rate (eGFR, mL/min/1.73 m^2^) was calculated using the abbreviated Modification of Diet in Renal Disease Study equation, corrected to a body surface area of 1.73 m^2^ [56]. The total lymphocyte count (TLC) was calculated as the WBC count (/mm^3^) × the percentage of lymphocytes in the blood. The total neutrophil count (TNC) was calculated as the WBC count (/mm^3^) × the percentage of neutrophils in the blood. The total monocyte count (TMC) was calculated as the WBC count (/mm^3^) × the percentage of monocytes in the blood.

Malnutrition status was considered for BMI < 18.5 kg/m^2^, albumin < 3.5 g/dL, TLC < 1.5 × 10^3^ cells/mm^3^, or when the patient-generated subjective global assessment (PG-SGA) scores ranged from 0 to 35: no malnourished status with scores of 0–3, moderately malnourished status with scores of 4–8, and severely malnourished status with scores ≥9 [57,58,59].

### 2.7. Body Composition Measurements

Body composition parameters—including LBM, TFM, and appendicular skeletal mass (ASM, including arms and legs)—were obtained by dual-energy fan-beam X-ray absorptiometry (Lunar iDXA, GE Medical Systems, Madison, WI, USA), following the guidelines of the International Society for Clinical Densitometry to accurately place each participant [60]. According to BMI and body size, the scan mode (standard, thin, or thick) was selected using scanner software. Scans were analyzed using the enCORE Software, version 15 (GE Lunar). Data were collected within 1 week before CCRT and within 3 days after CCRT.

### 2.8. Ultrahigh-Performance Liquid Chromatography (UPLC)-Based Measurements

Serum histidine, leucine, ornithine, and phenylalanine levels were measured as previously described [49,51,61]. Briefly, EDTA-treated plasma samples were collected within 1 week before CCRT and 3 days after CCRT. They were stored at −80 °C until assayed. Plasma samples (100 μL) were precipitated by adding an equal volume of 10% sulfosalicylic acid containing 200 μM norvaline (an internal standard). After protein precipitation and centrifugation at 12,000× *g* for 10 min at room temperature, derivatization was initiated by adding 10 mM AQC in acetonitrile. Eluent A (20 mM ammonium formate/1.0% acetonitrile) was added to the mixture after 10 min of incubation, and the amino acids were analyzed using the ACQUITY UPLC System [62,63], which comprised a binary solvent manager, sample manager, and tunable UV detector. The system was controlled and data were collected using Empower™ 2 Software. Separations were performed on a 2.1 × 100 mm ACQUITY BEH C18 column at a flow rate of 0.70 mL/min. The average intra-assay coefficients of variation for histidine, ornithine, leucine, and phenylalanine were 4.3%, 4.6%, 4.5%, and 4.6%, respectively. The total coefficients of variation for histidine, ornithine, leucine, and phenylalanine were 3.1%, 3.6%, 4.1%, and 3.7%, respectively. The detection limit was 0.5 μM for histidine, 2.0 μM for ornithine, 0.9 μM for leucine, and 3.3 μM for phenylalanine. The linear range was 25–500 μM for these four amino acids.

∆ indicates the interval changes in the abovementioned biochemical and anthropometric data, blood NIBs, DXA-derived parameters, and four metabolites before and after the CCRT course.

### 2.9. Statistical Analysis

SPSS (version 22.0; SPSS Inc., Chicago, IL, USA) was used for statistical analyses. Based on a power of 80%, α error of 0.05, and the annual number of patients with LAHNSCC receiving CCRT at our institute, the calculated minimum sample size was 42. Considering the drop-out rate due to CCRT toxicity or poor compliance with the data collection schedule, we assumed that the attrition rate was 30% and the required sample size of patients was 50. All continuous variables were examined for normality before the analysis. Independent *t*-tests or nonparametric Mann–Whitney tests were used for continuous variables, where appropriate. The chi-squared test was used for categorical variables. Paired *t*-tests and the Wilcoxon signed-rank test were used to detect differences before and after CCRT.

The association of different clinicopathological variables, biochemical and anthropometric data, NIBs, DXA-derived parameters, nutritional and calorie supply, metabolites, and CCRT factors with treatment interruption events was analyzed using logistic regression analysis. Forward stepwise selection was used in univariate and multivariate analyses. All independent variables that were significantly associated with treatment interruption events (*p* < 0.05) in the univariate analysis were included in the multivariate analysis.

Receiver operating characteristic (ROC) curves were used to determine the optimal cutoff value of treatment-interval changes in albumin and histidine levels, both of which were statistically significant in multivariate analysis. All differences in mortality rates were considered to be statistically significant (two-tailed *p*-value < 0.05).

The univariate associations of different clinicopathological variables, biochemical and anthropometric data, NIBs, DXA-derived parameters, nutritional and calorie supply, metabolites, and CCRT factors with treatment-interval changes in albumin and histidine levels were first analyzed using simple linear regression, nonparametric Mann–Whitney tests, or Kruskal–Wallis tests. All independent variables that were significantly associated with treatment-interval changes in albumin and histidine levels (*p* < 0.05) in the univariate analysis were included in the multivariable linear regression model analysis with forward stepwise selection. Variance inflation factors were used as variables to test collinearity.

Correlation matrices used to visualize correlations in treatment-interval changes among metabolites, biochemical and anthropometric factors, NIBs, and DXA-derived parameters were obtained using the Pearson correlation coefficient between each pair of variables, and were constructed using Statgraphics Centurion version 19 (Statgraphics Technologies, Inc., The Plains, VA, USA).

## 3. Results

Fifty patients were included in this study. Two patients requested to drop out because their families moved to other counties. Four patients completed the CCRT course with no treatment interruption events but were classified into the “incomplete CCRT group” because they completed the required tests until one week after CCRT. Forty-four patients were eligible for analysis at the end of the study. The recruitment, allocation, treatment modalities, and data collection schedule are shown in Figure 1. The clinical features of patients with LAHNSCC who completed CCRT are presented in Table 1.

### 3.1. Treatment-Interval Changes of the Anthropometric Data, NIBs, DXA-Related Measurements, and Serum Metabolites in Patients with LAHNSCC Completing CCRT

Of the 44 patients with LAHNSCC, 25 with oral cavity cancer received postoperative adjuvant CCRT, and 19 with non-oral-cavity cancer received primary CCRT. The study participants were predominantly men (95.5%), and had an average age of 54.7 years. The tongue (27.2%) was the most common tumor site, followed by the hypopharynx (22.7%) and the gingiva (11.4%). Nearly 90% of the tumors belonged to non-metastatic TNM stage IV, and all had at least one comorbid illness. Most patients were exposed to smoking (84.1%), alcohol consumption (72.7%), and betel nut usage (63.6%); they had an advanced tumor size (T3 + T4:75.0%), extended regional lymph node invasion (N2 + N3:65.9%), good histological differentiation (well + moderate: 86.4%), and optimal ECOG performance status (0 + 1:95.5%); 40% of the patients with cancer underwent tracheostomy before CCRT. Malnutrition rates were assessed using different malnutrition criteria before CCRT: PG-SGA-defined severe malnourished status (20.5%), BMI <18.5 kg/m^2^ (22.7%), albumin levels <3.5 g/dL (11.4%), and TLC <1.5 × 10^3^ cells/mm^3^ (31.8%) (Table 1).

Throughout the treatment course. patients consumed 26.9 ± 6.6 kcal/kg/day, which included intake of 3.7 ± 0.9 g of carbohydrates/kg/day, 1.0 ± 0.3 g of protein/kg/day, and 0.8 ± 0.2 g of lipids/kg/day. Nearly 30% of the patients had a calorie intake of ≥30 kcal/kg/day. Half of the patients underwent feeding tube placement, with a mean duration of 20.3 days, for calorie and protein supply (Table 1). Under the recommended calorie and protein support, patients showed significant reductions in BW, BMI, eGFR, fasting sugar, Hb levels, WBC count, TLC, triglyceride levels, all DXA-derived measurements, and the levels of two amino acid metabolites (leucine and ornithine), along with increased TNC and CRP levels. There were no interval changes in hepatic function, TMC, or levels of cholesterol, uric acid, albumin, prealbumin, transferrin, histidine, and phenylalanine at the end of CCRT. Eighteen patients (40.9%) experienced grade 3/4 non-hematologic toxicity, the most common form being infection (31.8%), followed by mucositis (25.0%). Seventeen patients (38.6%) developed grade 3/4 hematological toxicity, with the most common form being neutropenia (34.1%), followed by thrombocytopenia (13.6%) (Table 2).

### 3.2. Association of Treatment Interruption and Interval Changes in Serum Albumin and Histidine Concentrations of Patients with LAHNSCC during CCRT

No patient in this cohort experienced interruptions from machine faults or maintenance during the treatment course.

Nine patients (20.4%) experienced treatment interruption during CCRT: four patients with RT break > 5 days alone, and five with both RT and chemotherapy breaks. The interruptions were due to grade 3/4 treatment-related toxicities (Table 3). Among these nine patients, the mean interruption duration was 9.3 ± 3.3 days (range, 6–16 days); the interruptions happened in the second half of the treatment course.

Considering patients with treatment interruption and those with no treatment interruption, there were no differences in the clinicopathological variables (i.e., age, sex, tumor location, tumor stage, status of T and N, histological differentiation, exposure to smoking, alcohol and betel nut usage, performance status, presence of tracheostomy, and PG-SGA score), mean daily intake (calories; consumption of carbohydrates, protein, and fat), the percentage and mean days of feeding tube placement during CCRT, pretreatment levels of variables including biochemistry and anthropometry, NIBs, DXA-related measurements (LBM, TFM, and ASM), and three individual metabolites (leucine, ornithine, and phenylalanine). Compared to the subgroup with no treatment interruption, the subgroup with treatment interruption presented lower CRP levels but higher histidine concentrations before CCRT, experienced longer RT duration, developed more treatment-interval decreases in uric acid, Hb, albumin, ASM, and histidine, and had a higher proportion of patients with grade 3/4 mucositis toxicity. Patients with treatment interruption experienced more grade 3/4 toxicities than those with no interruption (Table 1).

Furthermore, there were significant correlations between treatment-interval levels of biochemical and anthropometric variables, NIBs, DXA-related measurements, and serum metabolites (Figure 2). The four amino acid metabolites and Hb were positively correlated, except for ornithine and phenylalanine. All treatment-interval changes in BW, BMI, and DXA-associated measurements were also positively correlated with one another, except for TFM with LBM and ASM. ASM changes were positively correlated with changes in transferrin, albumin, and histidine levels. The changes in albumin levels were positively correlated with those in prealbumin, transferrin, uric acid, and cholesterol levels, as well as ASM. CRP changes were negatively correlated with changes in histidine and uric acid levels. This correlation analysis suggested that intricate relationships between treatment-interval changes in these blood variables, anthropometric data, and body composition parameters were present, and could cause the occurrence of treatment interruption.

We performed multivariate analysis after adjustment for all covariates, including clinicopathological variables, biochemical and anthropometric data, mean daily intake, feeding tube placement, CCRT-related factors, blood NIBs, body composition parameters, and four individual amino acid metabolites. Only two variables—i.e., treatment-interval changes in albumin and histidine levels—were independently correlated with treatment interruption over the CCRT course (Table 4). Patients with higher albumin and histidine loss during CCRT had a greater treatment interruption rate (Figure 3). Nonetheless, energy consumption of ≥30 kcal/kg/day, duration and percentage of feeding tube placement, grade 3/4 mucositis toxicity, and more grade 3/4 toxicities were not correlated with treatment interruption (Table 4).

### 3.3. Factors Associated with Treatment-Interval Changes in Albumin and Histidine Levels of Patients with LAHNSCC Completing CCRT

We further investigated whether clinicopathological status, pretreatment nutritional and inflammatory conditions, treatment modalities, and relevant adverse effects of CCRT could affect the treatment-interval changes in albumin and histidine levels in patients with LAHNSCC completing CCRT. In multivariate analysis, the interval change in albumin levels was positively correlated with those in transferrin and uric acid levels; the treatment-interval change in histidine levels was positively correlated with phenylalanine level changes, but negatively correlated with CRP level changes (Table 5). Nonetheless, all variables regarding clinicopathological status, pretreatment nutritional and inflammatory conditions, and treatment modalities failed to show independent relationships with treatment-interval changes in serum albumin and histidine levels (Table 5).

## 4. Discussion

To overcome the confounding effects from mixed stages, varied treatment plans, and inadequate energy and protein supply during the treatment course (which possibly caused treatment interruptions), we prospectively analyzed a homogeneous group of patients with LAHNSCC undergoing the same standard CCRT under the provision of recommended calories of at least ≥25 kcal/kg/day and protein of 1.0 g/kg/day. Compared with treatment completion and toxicity profiles reported from previous studies [9,10,12,13,14,15,16,17,18], our results showed that 44 patients (88.0%) completed the current CCRT schedule and all required examinations with no delay; <40% patients experienced grade 3/4 treatment-related toxicities, and approximately one-fifth of patients experienced treatment interruptions, indicating that following the current study protocol, nearly 90% of patients with LAHNSCC were able to complete the CCRT course within the planned treatment duration, and fewer patients succumbed to treatment-related toxicities and interruptions. Therefore, comprehensive and multidisciplinary teamwork—including administrative work coordination, suitable delivery schedule of CCRT, psychological and social support, and adequate supply of energy and protein—could prevent unnecessary drop-outs and non-compliance events, improve treatment completion rates, and reduce treatment-induced adverse effects and interruptions.

Acute toxicity from CCRT needs further research, and remains a major limitation of continuous and uninterrupted CCRT courses [16,17,18]. In particular, grade 3/4 mucositis was reported as the root cause of unplanned treatment interruption in patients with HNC undergoing RT or CCRT, because mucositis is often accompanied by pain, aspiration, inanition, and infection, leading to discontinuation of the treatment course to allow healing, but negatively impacting prognosis [64,65,66]. The incidence rate of mucositis toxicity was 39% in patients with RT and 43% in patients with CCRT [9], accounting for 11–47% of interruption events during the treatment course [65,66]. Patients with LAHNSCC showed a high prevalence of malnutrition at the time of diagnosis, and developed malnutrition during the treatment course [67]. Malnutrition is a risk factor for the development of mucositis [68]; therefore, appropriate nutritional intervention during the CCRT course (to reduce the incidence of mucositis and other toxicities, as well as the resultant treatment interruption) is a reasonable approach. Nonetheless, our observations regarding the interplay between nutritional status, mucositis toxicity, and interruption events among patients with HNC undergoing CCRT remain inconsistent with those of previous studies. Some investigations showed that patients with optimal nutritional status suffer less from mucositis toxicity [69,70], and those with less incidence of treatment-related toxicity experience fewer treatment interruption events [9,71,72,73,74]. Russo et al. reviewed the relationship between RT breaks and oral mucositis, and reported that nutritional counseling and diet therapy—such as nutritional supplements and feeding tube placement—could reduce mucositis toxicity and the resulting treatment interruption in patients with HNC undergoing RT or CCRT [9]. Paccagnella et al. reported that HNC patients undergoing CCRT and early nutritional intervention had a lower incidence of RT breaks >5 days than those receiving standard care [73]. Lewis et al. further showed that patients with LAHNSCC undergoing CCRT who received prophylactic tube placement had a higher rate of chemotherapy completion than those who did not undergo tube placement [72]. In contrast, some studies showed that nutrition counseling or diet therapy—such as nutritional supplements and feeding tube placement—in patients treated with CCRT may have no benefit in reducing the frequency of treatment interruption events [69,74]. Moreover, the application of oral nutritional supplements is not advantageous in treatment interruption in patients with HNC receiving RT or CCRT [75], and there is no difference in the treatment interruption rate of patients with HNC receiving CCRT under nutritional supply via feeding tube placement [76,77,78,79]. These discrepancies among studies could be attributed to the inevitable bias of the retrospective study design, heterogeneous enrollment, and varied nutritional approaches. Thus, the present study, with a prospective design, shows essential observations. First, patients with treatment interruptions received fewer chemotherapy doses and developed a longer RT duration than those without interruptions. Furthermore, treatment interruption had no association with clinicopathological variables, pretreatment data pertaining to blood cell count, biochemical function, anthropometric measurements, NIBs and HLOP metabolites, body composition, calorie intake (≥30 kcal/kg/day), or the duration and percentage of feeding tube placement. In contrast to the pretreatment variables, significant decreases in treatment-interval levels of several variables—including hemoglobin, uric acid, albumin, ASM, and histidine metabolites—were detected in patients with treatment interruptions. Additionally, there was a strong correlation between treatment-interval changes in these blood variables and body composition parameters. Lastly, grade 3/4 mucositis and the number of grade 3/4 toxicities remained the risk factors with significant differences between the treatment-interruption and no-treatment-interruption groups. Thus, the treatment-interval level changes in certain variables—such as hemoglobin, uric acid, albumin, ASM, and histidine—could significantly affect the relationship between these acute morbidities and treatment interruption.

Acute toxicity from CCRT needs further research, and remains a major limitation of continuous and uninterrupted CCRT courses [16,17,18]. In particular, grade 3/4 mucositis was reported as the root cause of unplanned treatment interruption in patients with HNC undergoing RT or CCRT, because mucositis is often accompanied by pain, aspiration, inanition, and infection, leading to discontinuation of the treatment course to allow healing, but negatively impacting prognosis [64,65,66]. The incidence rate of mucositis toxicity was 39% in patients with RT and 43% in patients with CCRT [9], accounting for 11–47% of interruption events during the treatment course [65,66]. Patients with LAHNSCC showed a high prevalence of malnutrition at the time of diagnosis, and developed malnutrition during the treatment course [67]. Malnutrition is a risk factor for the development of mucositis [68]; therefore, appropriate nutritional intervention during the CCRT course (to reduce the incidence of mucositis and other toxicities, as well as the resultant treatment interruption) is a reasonable approach. Nonetheless, our observations regarding the interplay between nutritional status, mucositis toxicity, and interruption events among patients with HNC undergoing CCRT remain inconsistent with those of previous studies. Some investigations showed that patients with optimal nutritional status suffer less from mucositis toxicity [69,70], and those with less incidence of treatment-related toxicity experience fewer treatment interruption events [9,71,72,73,74]. Russo et al. reviewed the relationship between RT breaks and oral mucositis, and reported that nutritional counseling and diet therapy—such as nutritional supplements and feeding tube placement—could reduce mucositis toxicity and the resulting treatment interruption in patients with HNC undergoing RT or CCRT [9]. Paccagnella et al. reported that HNC patients undergoing CCRT and early nutritional intervention had a lower incidence of RT breaks >5 days than those receiving standard care [73]. Lewis et al. further showed that patients with LAHNSCC undergoing CCRT who received prophylactic tube placement had a higher rate of chemotherapy completion than those who did not undergo tube placement [72]. In contrast, some studies showed that nutrition counseling or diet therapy—such as nutritional supplements and feeding tube placement—in patients treated with CCRT may have no benefit in reducing the frequency of treatment interruption events [69,74]. Moreover, the application of oral nutritional supplements is not advantageous in treatment interruption in patients with HNC receiving RT or CCRT [75], and there is no difference in the treatment interruption rate of patients with HNC receiving CCRT under nutritional supply via feeding tube placement [76,77,78,79]. These discrepancies among studies could be attributed to the inevitable bias of the retrospective study design, heterogeneous enrollment, and varied nutritional approaches. Thus, the present study, with a prospective design, shows essential observations. First, patients with treatment interruptions received fewer chemotherapy doses and developed a longer RT duration than those without interruptions. Furthermore, treatment interruption had no association with clinicopathological variables, pretreatment data pertaining to blood cell count, biochemical function, anthropometric measurements, NIBs and HLOP metabolites, body composition, calorie intake (≥30 kcal/kg/day), or the duration and percentage of feeding tube placement. In contrast to the pretreatment variables, significant decreases in treatment-interval levels of several variables—including hemoglobin, uric acid, albumin, ASM, and histidine metabolites—were detected in patients with treatment interruptions. Additionally, there was a strong correlation between treatment-interval changes in these blood variables and body composition parameters. Lastly, grade 3/4 mucositis and the number of grade 3/4 toxicities remained the risk factors with significant differences between the treatment-interruption and no-treatment-interruption groups. Thus, the treatment-interval level changes in certain variables—such as hemoglobin, uric acid, albumin, ASM, and histidine—could significantly affect the relationship between these acute morbidities and treatment interruption.

In multivariate analysis, we further elucidated that toxicity was not correlated with treatment interruption, i.e., interval changes in albumin and histidine levels during CCRT were independently and positively correlated with treatment interruption (Table 4). Furthermore, the change in albumin levels was positively correlated with the interval changes of transferrin and uric acid; the histidine change was positively correlated with the change in phenylalanine levels, but negatively correlated with the change in CRP levels (Table 5). The current analysis suggests that toxicity could appear to cause treatment interruption, but the underlying nutritional and/or inflammatory changes during CCRT may be responsible for the treatment interruption. Thus, the main factors behind alterations in nutrition and inflammation, mucositis toxicity, and treatment interruption could be substances with antioxidative, anti-inflammatory, and metal-ion-chelating effects in the blood. Serum albumin has multiple ligand-binding capacities and free-radical-scavenging properties; thus, it exerts antioxidant and anti-inflammatory functions, accounting for over 80% of the antioxidant activity in the blood [80,81]. Compared to other blood proteins, it is primarily exposed to reactive oxygen species (ROS) because of its free thiol group of the Cys34 residue [80,81]; it is the main extracellular molecule responsible for modulating the plasma redox state, augmenting intracellular glutathione levels, and regulating cell signaling through the ubiquitous transcription factor nuclear factor kappa B (NF-κB), which mediates pro-inflammatory stress [82,83,84]. The antioxidative and anti-inflammatory functions of serum albumin explain the pathological process, and the albumin supply improves the clinical condition in certain critical and chronically ill patients [84,85,86,87,88,89,90]. Ishizuka et al. also reported a strong correlation between serum levels and neutropenia toxicity in patients with HNC receiving cisplatin-based CCRT, and emphasized that decreased binding activity of albumin to metal ions at low serum albumin concentrations could result in increased cisplatin-related toxicity [6,81]. Furthermore, serum uric acid could serve as a free radical scavenger in the blood and perform normal-range antioxidative functions to protect erythrocytes and immune cells [91]. Chang et al. found that low pretreatment uric acid (<5.05 mg/dL) was associated with lower survival outcomes in Taiwanese patients with oral cavity squamous-cell carcinoma undergoing surgery [92]. Lin et al. reported that high post-treatment uric acid (>5.06 mg/dL) was correlated with better survival outcomes in Chinese patients with nasopharyngeal carcinoma undergoing RT or CCRT [93]. Additionally, serum transferrin performs its antioxidative function by binding with iron in an inactive redox form, because free iron ions can stimulate the production of free radicals [83]. Transferrin expression is also affected by inflammation and malnutrition in the blood [94,95]. Histidine—a dietary essential amino acid, and one of the least abundant amino acids in the human body—is commonly recognized as one of the most active and antioxidative amino acids [96,97]. It acts as a scavenger of the hydroxyl radicals produced by the metal-ion-mediated redox reaction and singlet oxygen via direct interactions with the histidine imidazole ring [98]. Histidine also acts as an anti-inflammatory agent by abolishing NF-κB-mediated production of pro-inflammatory cytokines [99] or downregulating prostaglandin E2 function by interacting with its imidazole ring [100]. The histidine residue in proteins of the body efficiently binds to cisplatin and carboplatin [101]. Finally, the relationship between histidine and phenylalanine is intimate in both molecular and clinical respects. A decrease in plasma histidine levels results in reduced phenylalanine oxidation in healthy individuals following a long-term low-histidine diet [102]. Changes in histidine and phenylalanine levels in patients with heart failure can be used to assess their responses to medications and nutritional interventions [49]. The transport of histidine and phenylalanine into cells requires L-type amino acid transporter 1 (LAT1), i.e., a membrane transporter responsible for transporting bulky amino acids, which is regulated by glutathione status and oxidative stress [103,104]. LAT1 is overexpressed in patients with oral cavity squamous-cell carcinoma [105]. Although the antioxidative ability of phenylalanine is not obvious [97], phenylalanine hydroxylase (PAH)—which is mainly present in the liver, and is active in other tissues such as the kidneys, brain, and pancreas—can quickly metabolize phenylalanine to tyrosine in healthy individuals [106]. Tyrosine is an amino acid with powerful antioxidative ability [97]. Thus, changes in the levels of certain proteins and amino acids in the blood during CCRT may provide a possible explanation for the relationship between mucositis toxicity and treatment interruption in LAHSCC patients undergoing CCRT.

Oxidative and inflammatory stresses—such as ROS and pro-inflammatory cytokines—induced by CCRT can potentially inflict harm on epithelial cells as well as other organs, including the liver, skin, muscle, and peripheral vascular tissue [69]. Subsequently, the body’s protective response prevents damage progression [83]. Antioxidants in the blood are important defensive and reparative systems of the human body. Thus, the induction of oxidative and inflammatory stress from CCRT could cause the consumption of serum antioxidants during the treatment course. Changes in serum levels of antioxidants may correspond to the magnitude of redox potential and inflammation caused by CCRT. Based on the increase or decrease in the CRP status over the CCRT course, we further analyzed the patients, because CRP’s sensitivity, specificity, and reproducibility are widely used to assess the severity of the systemic inflammatory response [107]; we found that all cases with treatment interruption were those of the increased-CRP group during CCRT. Meanwhile, the increased-CRP group revealed more treatment-interval losses in albumin, prealbumin, transferrin, uric acid, and histidine levels, along with LBM and ASM, and a higher proportion of mucositis toxicity (Appendix A). Although further research is required, these analyses raise a hypothetical pathogenesis to decipher the relationship between nutritional–inflammatory changes and treatment toxicity and interruptions in patients with LAHNSCC undergoing CCRT: Serum albumin neutralizes the oxidative and inflammatory stresses from CCRT. When serum albumin levels fluctuate during the treatment course, serum proteins and metabolites such as transferrin and uric acid are utilized to maintain the optimal plasma antioxidative and anti-inflammatory function against CCRT (Table 5). Simultaneously, serum histidine counteracts CCRT-induced oxidative and inflammatory damage. The decrease in serum histidine levels might arouse the influx of serum phenylalanine into cells via LAT1, resulting in reduced phenylalanine levels. Within the cell, phenylalanine hydroxylase catalyzes phenylalanine to tyrosine, which performs antioxidative functions intracellularly and extracellularly [108,109]. When both serum albumin and histidine are spent and faded away, the muscle mass can carry out proteolysis and release albumin and histidine into the blood [110,111], while hemoglobin (another storehouse of histidine in the body) can also be degraded to maintain serum histidine levels [102,112,113].

In the present study, patients with treatment interruptions may have developed more oxidative and inflammatory stress than those with no interruptions, leading to increased consumption of albumin and histidine, and subsequent muscle proteolysis and Hb breakdown. Although the release of albumin and histidine from the muscle and Hb was to compensate for the rapid loss of antioxidants in the blood, and to avoid the development of toxicity, the outcome was unsuccessful—possibly due to excessive oxidative and inflammatory damage. Accordingly, a reduced antioxidant pool could not protect the integrity and function of the mucosa and organs from toxicity. Once toxicities develop, they can deteriorate the status of oxidative and inflammatory insults from CCRT and make themselves worse. Finally, treatment interruption takes place in this vicious cycle (Figure 4).

The strengths of the present study are its homogeneous patient population treated with the standard CCRT protocol under the provision of recommended calories and protein over the treatment course, along with the simultaneous analysis of all confounding covariates. However, further research is needed for more detailed understanding. First, because patients with HNC undergoing CCRT require over 55,000 calories to complete the treatment course [114], the albumin and histidine released from the muscle mass and hemoglobin may be metabolized for energy fuel. For instance, histidine can be converted to glutamate, which can enter the glutamate–ornithine–proline pathway or the tricarboxylic acid cycle to produce energy [49]. In the present study, the mean BW was approximately 60 kg, and the average treatment duration was 42 days. Under the mandatory provision of at least 25 kcal/kg/day of energy, the patient would receive more than 60,000 calories throughout the entire CCRT course. The conversion of proteins and histidine to energy is trivial. Second, the changes in serum levels of albumin reflect the dynamic state between albumin synthesis and catabolism, including degradation and clearance in response to stress, inflammation, and diseases [115]. Fearon et al. demonstrated that the albumin synthesis rate is not changed in cancer patients even with low serum albumin concentrations [116]. Our results reported no grade 3/4 diarrhea toxicity, and there were no significant differences in interval changes regarding liver function, renal clearance (ΔALT, Δtotal bilirubin, and ΔeGFR), or the intake of calories, carbohydrates, proteins, and fat between the interruption and no-interruption groups. It is thus less likely that insufficient synthesis, increased gastrointestinal loss, and renal clearance could affect the interval changes in albumin levels during CCRT and lead to the resultant treatment interruption. Third, the decrease in treatment-interval serum histidine levels could be a consequence of enhanced histidine catabolism [117] or its increased accumulation in the muscle, where histidine is converted into strong antioxidative dipeptides such as carnosine [118] to cope with CCRT-induced oxidative stress causing muscle breakdown. Histidine metabolism is closely related to food and energy intake [117], and plasma histidine levels are positively correlated with protein synthesis [102]. Our data showed that under adequate energy and protein supply, there were no differences in histidine levels before and after CCRT (Table 2), no differences in the consumption of energy and protein between the interruption and no-interruption groups (Table 1), and a positive correlation between changes in histidine levels and muscle mass (Figure 2). This supports the hypothesis that serum histidine is consumed under oxidative and inflammatory stress, followed by muscle breakdown to supplement histidine loss, rather than being used to enhance catabolism or carnosine synthesis during CCRT. Additionally, the postulate pertaining to the PAH conversion of phenylalanine into tyrosine—which is assumed to be an antioxidant backup for histidine during CCRT—is debatable, because the activity of PAH could be impaired, and phenylalanine may produce abnormal tyrosine isomers that disrupt cellular homeostasis via oxidative and inflammatory stress from cancer and radiation [119,120]. However, Davis et al. analyzed the relationship between the structure and function of PAH under radiation exposure, and found that the activity of PAH might not be altered under regulation by the tetrahydrobiopterin cofactor [121,122]. Hence, the postulate could be true, and radiation exposure might modulate PAH function without completely impairing its ability to convert to tyrosine, because the provision of adequate calories and proteins throughout the entire CCRT course could replenish the consumption of the tetrahydrobiopterin cofactor during CCRT and, subsequently, maintain PAH function. Finally, leucine and ornithine concentrations decreased during CCRT. Leucine, an essential branched-chain amino acid, is among the most abundant amino acids in proteins. It stimulates protein synthesis via the mTOR signaling pathway, and inhibits proteolysis by downregulating the ubiquitin–proteasome proteolytic pathway [123,124]. Leucine can be obtained via protein catabolism from diet or from the breakdown of tissue protein stores. It acts as the major nitrogen donor to build common body proteins in tissues such as the skeletal muscle, liver, and pancreas [125,126]. Leucine, like other branched-chain amino acids, can reduce oxidative stress in chronic illnesses [127]. When muscles are exposed to chemotherapy or radiotherapy, muscle proteins undergo proteolysis, mainly via ubiquitin-dependent NF-κB-mediated or ROS-induced cytokine (e.g., tumor necrosis factor-α, interleukin-1, and interferon-γ) pathways [128]. Leucine supplementation activates mTOR signaling, which is a master regulator of muscle mass and function [129]. Our findings showed that two variables (TLC and leucine) at pretreatment and two (hemoglobin and ornithine) during the treatment course were associated with treatment-interval changes in leucine levels (Appendix A). Thus, decreases in serum leucine levels during CCRT could be due to the shift of leucine from the blood to the intracellular microenvironment, where it repairs tissue protein damage from oxidative and inflammatory stress caused by CCRT. Pretreatment TLC was correlated with pretreatment levels of albumin (r = 0.32, *p* < 0.05), prealbumin (r = 0.40, *p* < 0.01), and ASM (r = 0.32, *p* < 0.05); these could represent the nutritional status. Pretreatment leucine levels are correlated with the total concentrations of all essential amino acids [130]. Both TLC and leucine pretreatment status may reflect the magnitude of nutritional status and leucine storage to cope with CCRT-induced tissue protein loss. Leucine is a major amino acid in hemoglobin, and it can improve anemia via mTOR signaling and iron metabolism [131,132]; thus, there could be a close correlation between leucine and hemoglobin during CCRT. The connection between leucine and ornithine during CCRT could be ascribed to the redistribution of leucine and other essential amino acids to maintain protein loss; it thus the influences the ornithine load that is converted from other amino acid pools. Ornithine, a nonessential amino acid, is an essential intermediate in the urea cycle, and plays an important role in the liver. The ornithine level is determined by the ornithine load generated from the metabolism of amino acids, clearance by the urea cycle, and ornithine catabolism—represented as the ornithine–spermine/spermidine (polyamine metabolite) pathway [133,134]. We found that only treatment-interval level changes in cholesterol were associated with those of ornithine (Appendix A); thus, ornithine catabolism (ornithine–polyamine pathway) could cause changes in serum ornithine levels during CCRT. Although the ornithine load could be affected because other amino acids were utilized to repair damaged proteins, patients received adequate energy and protein supply, and their hepatic functions (ALT and total bilirubin levels) were not altered during the CCRT course. The conversion of ornithine to citrulline in the urea cycle should be not significantly affected. Thus, the ornithine–polyamine pathway may explain the unusual association between ornithine and cholesterol. Oxidative stress can increase the activity of spermidine/spermine N^1^-acetyltransferase (SSAT)—the rate-limiting enzyme in polyamine catabolism [135]. Jell et al. showed an accumulation of body fat and an increase in acetyl- and malonyl-CoA pools in the white adipose tissue of SSAT-knockout mice [136]. Kramer et al. reported that increased SSAT activity augments metabolic flux via the polyamine pathway owing to enhanced ornithine decarboxylase function, which reduces polyamines via SSAT [133,137]. This vicious cycle results in the fall of acetyl-CoA and malonyl-CoA—i.e., precursors of cholesterol and fatty acids—and eventually results in a substantial increase in glucose oxidation and fat tissue loss [133]. Overexpression of SSAT could also reduce plasma cholesterol levels via the enhanced hepatic cholesterol 7α-hydroxylase activity, which converts cholesterol to bile acid in transgenic animal models [138]. Thus, CCRT-induced oxidative and inflammatory stress might be responsible for the reduction in leucine and ornithine levels during the treatment course.

Our results for LAHNSCC were partially consistent with those of previous reports on other malignancies [139,140,141,142,143]. Patients with esophageal cancer receiving CCRT who exhibited increased albumin levels during the treatment course had better survival outcomes [139,143]. Kayauchi et al. analyzed 226 patients with lung cancer undergoing cytotoxic chemotherapy, and reported that patients with decreased albumin levels during treatment significantly suffered from febrile neutropenia toxicity [141]. The histidine degradation pathway is essential for determining the chemosensitivity of tumor cells to methotrexate; thus, the addition of histidine to methotrexate significantly reduces tumor growth in animal models [140]. Exogenous histidine treatment has antitumor effects on liver cancer cell lines, and can reverse sorafenib resistance by modulating LAT1 [142]. Additionally, some reports have shown that supplementation with certain amino acids might improve protein synthesis in cancer with cachexia [38,144]. Although these reports—including ours—suggest that albumin and histidine supplementation during the anticancer treatment could be a potential and reasonable strategy in daily practice, its clinical application remains ambiguous. ROS, ionizing radicals, and inflammation generated from radiation and chemotherapy mainly eradicate cancer cells and cause toxicity during the treatment course; thus, it is unclear whether exogenous antioxidants such as albumin and histidine improve or limit the overall treatment outcomes. Additionally, growing evidence indicates that, clinically speaking, reduced serum albumin levels are a result rather than an etiology for diseases [115]. Furthermore, there is uncertainty regarding the outcomes of histidine supplementation at the molecular level, such as variations in gene expression and different metabolic fates of excess histidine in various tissues. These concerns may leave a considerable gap in the application of albumin and histidine.

This study had several limitations. Although treatment interruption had a negative impact on the survival outcomes of cancer patients receiving RT or CCRT [19,20,21,22], patients with no treatment interruptions showed a marginal benefit in survival outcomes compared to those with interruptions (6-month recurrence rate: 8.6% vs. 33.3%, *p* = 0.054; one-year mortality rate: 11.4% vs. 33.3%, *p* = 0.109; two-year mortality rate: 17.14% vs. 44.4%, *p* = 0.081) in this study. This could be ascribed to treatment interruption as the primary endpoint. Therefore, the sample size may not be sufficient for survival analysis. Second, the status of oxidation and inflammation before and after CCRT is not fully understood. Although we used treatment-interval CRP changes to reflect the dynamic state of oxidative and inflammatory stress, and found some significant correlations between CRP changes, treatment toxicity, and interruptions, further research is required. For instance, patients with increased CRP levels developed a similar number of grade 3/4 toxicities as those with decreased CRP levels during CCRT (Appendix A), but patients with interruptions developed more high-grade toxicities than those with no interruptions (Table 1). This discrepancy suggests that certain potential oxidative and inflammatory stress biomarkers—such as malondialdehyde, glutathione, and pro-inflammatory cytokines, which could correspond to the degree of oxidative and inflammatory stress—should be examined and used to reclassify the redox and inflammation status of patients. Furthermore, the enrolled patients were Taiwanese, and most of them were men (95.5%). The present results should be cautiously interpreted for different ethnic groups, females, varied treatment schedules, nutrition support plans, and regional susceptibility to this disease. Although the advantage of this study was a prospective design with homogeneous enrollment, standard protocol, and accurate sample size calculation that was based on the head and neck cancer registry of our institute, the issue that all patients were recruited from a single institution should be addressed. Because there might be the presence of selection bias, future investigation with a prospective large-scale and multi-institutional study is warranted. Additionally, a major weakness of this study was the lack of a comprehensive amino acid metabolite profile before and after CCRT. This drawback may leave the postulate regarding the conversion of phenylalanine into tyrosine up for debate. We believe that even if the entire amino acid panel is completed and histidine and phenylalanine are replaced, certain amino acids with antioxidative and anti-inflammatory functions can counteract CCRT-induced insults. The hypothesis regarding the contribution of serum albumin and amino acids against oxidative and inflammatory stress still holds. Lastly, although six patients failed to meet the criteria of the final analysis (two drop-outs and four with incomplete data collection during CCRT), there were no statistical differences in baseline characteristics between the complete-CCRT and incomplete-CCRT groups (Appendix A). Thus, the impact of drop-outs and incomplete data collection was not considered in the present analysis.

## 5. Conclusions

This prospective observational study demonstrated that among patients with LAHNSCC receiving CCRT under the recommended energy and protein supply (≥25 kcal/kg/day and 1.0 g/kg/day of protein) over the treatment course, treatment interruption was associated with treatment-interval changes in serum albumin and histidine levels, both of which likely reduce the oxidative and inflammatory stress generated from CCRT, as well as the resultant treatment-associated toxicity. Follow-up of serum albumin and histidine levels during CCRT may be considered as a routine nutritional assessment for these patients.

## Figures and Tables

**Figure 1 cancers-14-03112-f001:**
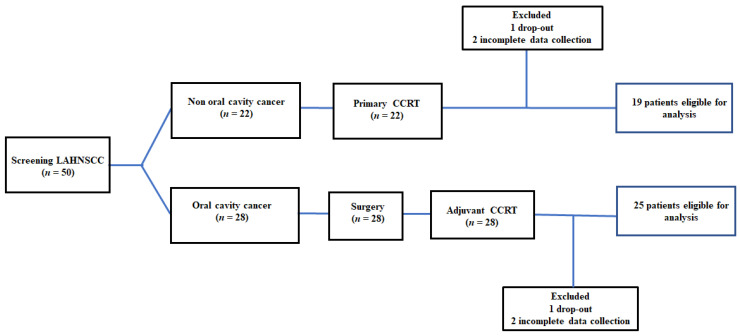
The flow diagram of recruitment: The incomplete CCRT group was defined as (1) patients who dropped out during the treatment course; and (2) incomplete data collection—patients who did not complete the required examinations or missed scheduled tests. LAHNC, locally advanced head and neck cancer; CCRT, concurrent chemoradiotherapy.

**Figure 2 cancers-14-03112-f002:**
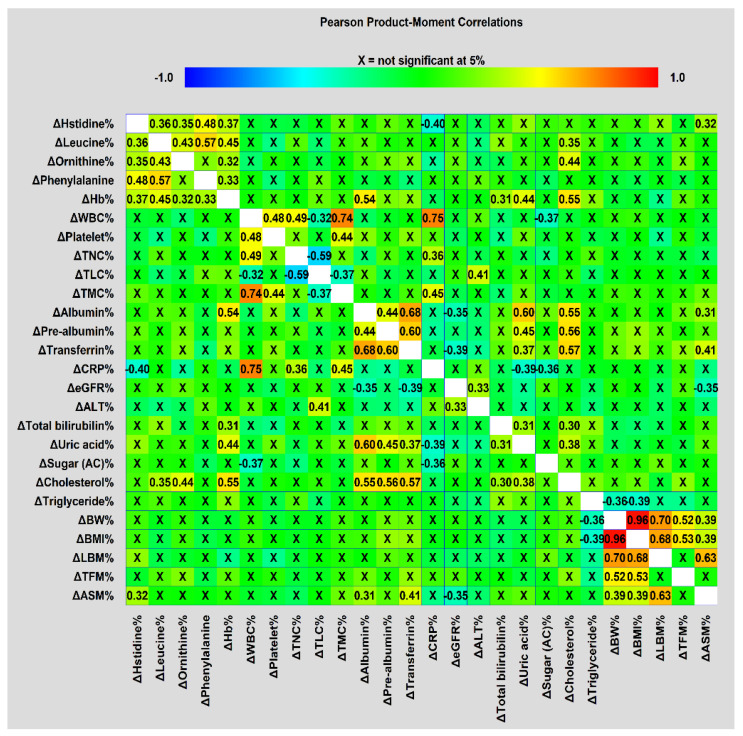
Correlation matrices used to visualize correlations in treatment-interval changes among metabolites, biochemical and anthropometric factors, NIBs, and DXA-derived parameters were obtained using Pearson’s correlation coefficient.

**Figure 3 cancers-14-03112-f003:**
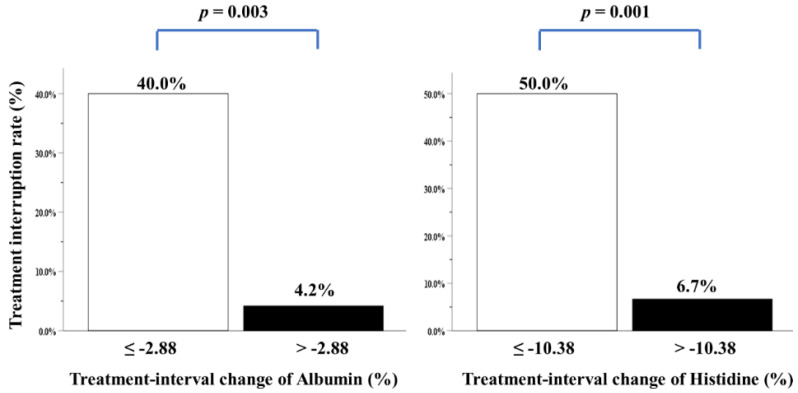
Treatment interruption rate of patients with LAHNSCC completing CCRT, stratified by treatment-interval changes in albumin or histidine levels. The cutoff value analyzed by ROC curves for treatment-interval changes in albumin levels was −2.88% (area under the curve (AUC): 0.762, *p* = 0.016), while that of treatment-interval changes in histidine levels was −10.38% (AUC: 0.768, *p* = 0.014).

**Figure 4 cancers-14-03112-f004:**
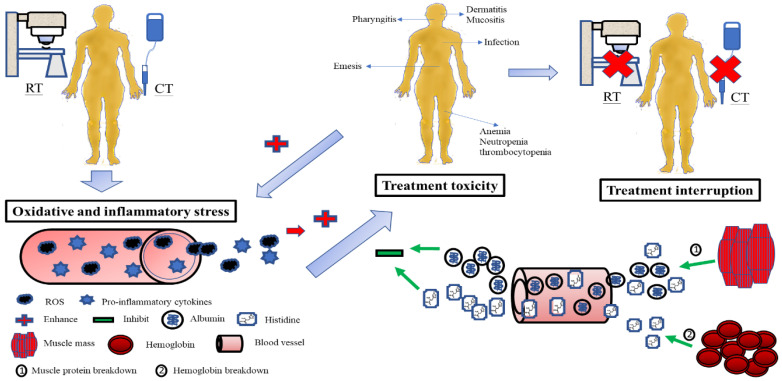
Possible pathogenesis explains the relationship between changes in serum albumin and histidine levels during CCRT, treatment toxicity, and treatment interruption in patients with LAHNSCC undergoing CCRT. The oxidative and inflammatory stress from CCRT consumes serum albumin and histidine. Subsequently, muscle proteolysis and hemoglobin breakdown release albumin and histidine into the blood to compensate for the serum antioxidant loss and stop the development of toxicity. If serum antioxidants such as albumin and histidine fail to neutralize oxidative and inflammatory stress, the integrity and function of the mucosa and organs are no more protected from toxicity damage. Treatment interruption inevitably takes place. RT, radiotherapy; CT, chemotherapy; CCRT, concurrent chemoradiotherapy; LAHNSCC, locally advanced head and neck squamous-cell carcinoma.

**Table 1 cancers-14-03112-t001:** Clinicopathological variables, biochemical and nutrition–inflammation data, anthropometric and body composition characteristics, serum HLOP metabolites, and treatment-related factors of 44 patients with LAHNSCC completing the CCRT course, stratified by treatment interruption.

Variables Expressed as Numbers (%) or Mean ± SD	All	Treatment with No Interruption	Treatment with Interruption	*p*-Value *
Included patient number	44 (100.0)	35 (79.5)	9 (20.5)	
*Clinicopathological factor*				
Age (years)	54.7 ± 9.2	55.1 ± 8.5	53.0 ± 11.8	0.543
≥65:<65	6 (13.6):38 (86.4)	5 (14.3):30 (85.7)	1 (11.1):8 (88.9)	0.805
Sex (male: female)	42 (95.5):2 (4.5)	24 (97.1):1 (2.9)	8 (88.9):1 (11.1)	0.289
Tumor location (oral cavity/other)				0.111
Buccal mucosa/tonsil	3 (6.8)/3 (6.8)	2 (5.7)/2 (5.7)	1 (11.1)/1 (11.1)	
Tongue/tongue base	12 (27.2)/1 (2.2)	11 (31.4)/1 (3.3)	1 (11.1)/0 (0.0)	
Gingiva/soft palate	5 (11.4)/0 (0.0)	5 (14.3)/0 (0.0)	0 (0.0)/0 (0.0)	
Retromolar/hypopharynx	2 (4.5)/10 (22.7)	1 (3.3)/7 (20.0)	1 (11.1)/3 (33.4)	
Hard palate/larynx	1 (2.2)/5 (6.7)	1 (3.3)/3 (8.6)	0 (0.0) 2 (22.2)	
Lip/--	2 (4.5)/--	2 (5.7)/--	0 (0.0)/--	
TNM stage				0.301
III:IVA:IVB	5 (11.4):25 (56.8):14 (31.8)	4 (11.4):18 (51.4):13 (37.1)	1 (11.1):6 (66.7):2 (22.2)	
T status				0.131
T0-2:T3-4	11 (25.0):33 (75.0)	7 (20.0):28 (80.0)	4 (44.4):5 (55.6)	
N status				0.400
N0-1:N2-3	15 (34.1):29 (65.9)	13 (37.1):22 (62.9)	2 (22.2):7 (77.8)	
Histological differentiation grade				
(well: moderately: poorly)	6 (13.6):32 (72.8):6 (13.6)	4 (13.6):28 (72.7):3 (13.6)	2 (22.2):4 (44.4):3 (33.4)	0.089
ECOG performance status (0:1:2)	3 (6.9):39 (88.6):2 (4.5)	1 (2.9):33 (94.2):1 (2.9)	2 (22.2):6 (66.7):1 (11.1)	0.066
Tracheostomy (no vs. yes)	27 (61.4):17 (38.6)	20 (57.1):15 (42.9)	7 (77.8):2 (22.2)	0.257
Smoking exposure (no vs. yes)	7 (15.9):37 (84.1)	5 (14.3):30 (85.7)	2 (22.2):7 (77.8)	0.562
Alcohol consumption (no vs. yes)	12 (27.3):32 (72.7)	9 (25.7):26 (74.3)	3 (33.3):6 (66.7)	0.647
Betel nut use (no vs. yes)	16 (36.4):28 (63.6)	12 (34.3):23 (65.7)	4 (44.4):5 (55.6)	0.572
HN-CCI (0:≥1)	0 (0.0):44 (100.0)	0 (0.0):35 (100.0)	0 (0.0):9 (100.0)	−−−−
PG-SGA assessment before CCRT				
Malnutrition (none: moderate: severe)	9 (20.5):26 (59.0):9 (20.5)	6 (17.1):22 (62.9):7 (20.0)	3 (33.3):4 (44.4):2 (22.2)	0.510
*Biochemical data*				
Before CCRT				
eGFR (mL/min/1.73 m^2^)	105.4 ± 28.8	105.2 ± 27.3	106.2 ± 36.1	0.928
ALT (U/L, normal ≤36)	22.5 ± 10.8	26.0 ± 15.6	19.4 ± 9.2	0.230
Total bilirubin (mg/dL, normal ≤1.3)	0.8 ± 0.4	0.4 ± 0.3	0.7 ± 0.7	0.327
Uric acid (mg/dL, normal <7.0)	5.2 ± 1.7	5.0 ± 1.7	5.9 ± 2.1	0.250
Sugar (AC, mg/dL)	115.2 ± 16.5	119.3 ± 52.2	99.3 ± 23.2	0.273
Treatment-interval change (%)				
∆eGFR% ⁑	−11.3 ± 26.7	−15.2 ± 28.0	−10.7 ± 26.3	0.873
∆ALT% ⁑	−22.5 ± 28.5	−28.0 ± 13.3	−7.2 ± 51.0	0.526
∆Total bilirubin% ⁑	−2.2 ± 8.5	−3.1 ± 10.4	0.8 ± 10.6	0.857
∆Uric acid% ⁑	−3.5 ± 46.7	2.1 ± 50.1	−25.7 ± 18.6	0.017 *
∆Sugar (AC) ⁑	−9.1 ± 36.7	−13.4 ± 38.4	−7.9 ± 22.8	0.120
*Anthropometric data and blood NIB data*				
Before CCRT				
BW (kg)	63.9 ± 12.4	64.4 ± 11.7	62.1 ±15.1	0.622
BMI (kg/m^2^)	23.1 ± 3.9	23.4 ± 3.8	22.0 ± 4.3	0.327
<18.5:≥18.5	10 (22.7):34 (77.3)	7 (20.0):28 (80.0)	3 (33.3):6 (66.7)	0.395
Hb (g/dL)	12.0 ±1.8	12.0 ±1.7	11.8 ± 2.3	0.812
WBC (×10^3^ cells/mm^3^)	6.6 ± 2.8	6.6 ± 2.8	5.6 ± 2.2	0.258
Platelet count (×10^3^/mm^3^)	260.1 ± 91.4	262.3 ± 91.0	251.8 ± 98.0	0.764
TLC (×10^3^ cells/mm^3^)	1.8 ± 0.6	1.9 ± 0.6	1.7 ± 0.8	0.464
<1.5:≥1.5	14 (31.8):30 (68.2)	10 (28.6):25 (71.4)	4 (44.4):5 (55.6)	0.362
TNC (×10^3^/mm^3^)	4.0 ± 0.3	4.2 ± 2.0.	3.3 ± 1.5.	0.254
TMC (×10^3^/mm^3^)	0.5 ± 0.3	0.5 ± 0.3	0.4 ± 0.1	0.379
Albumin (g/dL)	3.9 ±0.4.	3.9 ±0.4	3.8 ± 0.4	0.801
<3.5:≥3.5	5 (11.4):39 (88.6)	4 (11.4):31 (88.6)	1 (11.1):8 (88.9)	0.979
Prealbumin (g/dL, normal: 20–40)	25.3 ± 5.4	25.6 ± 4.9	24.3 ± 7.6	0.524
Transferrin (g/dL normal: 200–360)	208.0 ± 34.6	208.2 ± 28.1	207.5 ± 55.1	0.946
Total cholesterol (mg/dL, normal <200)	169.3 ± 41.8	168.9 ± 40.5	170.7 ± 49.1	0.923
Triglyceride (mg/dL, normal <150)	160.5 ± 15.4	164.1 ± 17.2	146.7 ± 37.1	0.655
CRP (mg/dL)	6.3 ± 7.7	7.4 ± 8.4	2.3 ± 3.1	0.003 *
Treatment-interval change (%)				
∆BW% ⁑	−4.9 ± 5.7	−4.8 ± 5.7	−5.4 ± 6.0	0.792
∆BMI% ⁑	−4.9 ± 6.4	−4.5 ± 6.3	−6.7 ± 7.0	0.358
∆Hb% ⁑	−13.3 ± 14.1	−11.8 ± 15.2	−18.9 ± 6.2	0.043 *
∆WBC% ⁑	−25.9 ± 51.6	−26.3 ± 50.4	−24.8 ± 59.6	0.938
∆Platelet% ⁑	−15.5 ± 37.8	−10.9 ± 34.6	−22.9 ± 46.2	0.121
∆TLC% ⁑	−24.8 ± 8.5	−21.6 ± 59.5	−37.5 ± 40.0	0.453
∆TNC% ⁑	16.4 ± 3.0	13.7 ± 18.4	17.3 ± 21.8	0.064
∆TMC% ⁑	−4.0 ± 10.9	−3.2 ± 12.8	−4.7 ± 17.8	0.189
∆Albumin% ⁑	−1.0 ± 14.1	2.4 ± 14.2	−10.0 ± 10.8	0.020 *
∆Prealbumin% ⁑	−9.7 ± 28.2	−9.5 ± 28.1	−11.5 ± 28.5	0.310
∆Transferrin% ⁑	−13.1 ± 13.3	−14.6 ± 20.2	−13.5 ± 23.5	0.824
∆Cholesterol% ⁑	0.9 ± 21.1	1.4 ± 22.1	−1.2 ± 17.6	0.738
∆Triglyceride% ⁑	−3.1 ± 41.6	−4.0 ± 42.6	0.4 ± 38.2	0.782
∆CRP% ⁑	609.5 ± 30.6	675.5 ± 31.9	599.8 ± 185.9	0.395
*DXA-related measurements*				
Before CCRT				
LBM (kg)	43.8 ± 5.9	44.1 ± 5.3	42.7 ± 8.3	0.533
TFM (kg)	17.5 ± 7.3	17.7 ± 7.5	16.8 ± 6.8	0.759
ASM (kg)	18.5 ± 3.3	18.7 ± 2.9	18.1 ± 4.7	0.614
Treatment-interval change (%)				
∆LBM% ⁑	−7.1 ± 6.0	−6.7 ± 6.8	−8.6 ± 4.1	0.421
∆TFM% ⁑	−3.4 ± 13.2	−2.7 ± 15.5	−6.1 ± 12.9	0.554
∆ASM% ⁑	−7.8 ± 8.5	−6.3 ± 7.2	−13.4 ± 7.2	0.027 *
*HLOP Metabolites*				
Before CCRT				
Histidine (μM)	78.8 ± 16.2	76.7 ± 16.0	87.1 ± 15.0	0.041 *
Leucine (μM)	128.7 ± 40.2	132.2 ± 40.9	115.3 ± 36.0	0.389
Ornithine (μM)	122.5 ± 35.4	118.2±30.8	139.1 ± 48.1	0.194
Phenylalanine (μM)	63.1 ± 10.4	62.8 ± 15.3	69.8 ± 20.5	0.174
Treatment-interval change (%)				
∆Histidine% ⁑	−1.4 ± 28.2	9.3 ± 26.2	−11.4 ± 31.2	0.017 *
∆Leucine% ⁑	−7.5 ± 27.2	−7.8 ± 28.0	−6.5 ± 25.4	0.897
∆Ornithine% ⁑	−7.0 ±23.4	−6.4 ± 23.2	−9.4 ± 25.0	0.738
∆Phenylalanine% ⁑	13.9 ± 33.3	15.0 ± 34.9	10.4 ± 27.5	0.675
Mean daily calorie intake during CCRT †	26.9 ± 6.6	26.8 ± 5.9	27.6 ± 8.9	0.741
≥25 and <30:≥30	31 (70.5):13 (29.5)	24 (68.6):11 (31.4)	7 (77.8):2 (22.2)	0.589
Mean daily protein intake during CCRT ††	1.0 ±0.3	1.0 ±0.2	1.0 ±0.3	0.872
Mean daily CHO intake during CCRT ††	3.7 ±0.9	3.7 ±0.8	3.9 ±1.3	0.588
Mean daily fat intake during CCRT ††	0.8 ± 0.2	0.9 ± 0.2	0.9 ± 0.3	0.775
Feeding tube placement (no vs. yes)	22 (50.0):22 (50.0)	17 (48.6):18 (51.4)	5 (55.68):4 (44.4)	0.709
Mean days of feeding tube placement during CCRT	20.3 ± 3.7	19.7 ± 3.8	22.1 ± 9.8	0.748
*CCRT data*				
Radiotherapy				
Dose (Gy)	66.6 ± 3.9	66.6 ± 3.5	66.8 ± 5.3	0.859
Fractions	32.6 ± 1.3	32.6 ± 1.1	33.4 ± 2.6	0.356
Duration (days)	48.4 ± 5.1	46.9 ± 2.6	55.0 ± 5.1	0.031 *
Cisplatin dose (mg/m^2^)	233.4 ± 22.2	238.1 ± 14.6	215.0± 35.7	0.004 *
Grade 3/4 toxicity during CCRT				
Non-hematologic				
Dermatitis (no vs. yes)	43 (97.7):1 (2.3)	34 (97.7):1 (2.3)	9 (100):0 (0.0)	0.608
Pharyngitis (no vs. yes)	42 (95.5):2 (4.5)	33 (94.3):2 (5.7)	9 (100):0 (0.0)	0.463
Infection (no vs. yes)	30 (68.2):14 (31.8)	26 (74.3):9 (25.7)	4 (44.4):5 (55.6)	0.089
Mucositis (no vs. yes)	33 (75.0):11 (25.0)	29 (82.9):6 (17.1)	4 (44.4):5 (55.6)	0.018 *
Emesis (no vs. yes)	41 (93.2):3 (6.8)	33 (93.2):2 (6.8)	8 (88.9):1 (11.1)	0.567
Hematologic				
Anemia (no vs. yes)	41 (93.2):3 (6.8)	33 (93.2):2 (6.8)	8 (88.9):1 (11.1)	0.567
Neutropenia (no vs. yes)	29 (65.9):15 (34.1)	25 (68.6):10 (31.4)	4 (44.4):5 (55.6)	0.463
Thrombocytopenia (no vs. yes)	38 (86.4):6 (13.6)	30 (85.7):5 (14.3)	8 (88.9):1 (11.1)	0.801
Number of grade 3/4 toxicities	1.2 ± 1.1	1.0 ± 1.1	2.1 ± 0.9	0.008 *

† Unit: Kcal/kg/day; †† unit: g/kg/day.* Comparing the difference between treatment interruption and no treatment interruption for each variable; *p* < 0.05, statistical significance. The Mann–Whitney test was used for ∆Hb%, CRP, ∆ uric acid%, total bilirubin, mean time for feeding tube placement, dose, fraction and days of radiotherapy, cisplatin dose, all DXA-related measurements, all metabolites, and the number of grade 3/4 toxicities. Independent *t*-tests were used for other continuous variables. The chi-squared test was used for all categorical data. ⁑ ∆ indicates the value obtained by subtracting the pretreatment value from the post-treatment value; % indicates (∆ value/the pretreatment value) ×100%. Abbreviations—LAHNSCC: locally advanced head and neck squamous-cell carcinoma; CCRT: concurrent chemoradiotherapy; SD: standard deviation; HN-CCI: Charlson Comorbidity Index; ECOG: Eastern Cooperative Oncology Group; CHO: carbohydrate; PG-SGA: patient-generated subjective global assessment; NIBs: nutrition–inflammation biomarkers; BW: body weight; BMI: body mass index; eGFR: estimated glomerular filtration rate; ALT: alanine aminotransferase; Hb: hemoglobin; WBC: white blood cell; TLC: total lymphocyte count; TNC: total neutrophil count; TMC: total monocyte count; CRP: C-reactive protein; DXA: dual-energy X-ray absorptiometry; LBM: lean body mass; TFM: total fat mass; ASM: appendicular skeletal mass; HLOP: histidine, leucine, ornithine, and phenylalanine.

**Table 2 cancers-14-03112-t002:** Treatment-interval changes of anthropometric data, NIBs, serum HLOP metabolites, and DXA-derived body composition measurements in 44 patients with LAHNSCC following CCRT completion.

Variables Expressed as Numbers (%) or Mean ± SD	CCRT Starts	CCRT Ends	*p*-Value *
** *Anthropometric data* **			
BW (kg)	63.9 ± 12.4	60.6 ± 11.6	<0.001
BMI (kg/m^2^)	23.1 ± 3.9	21.8 ± 3.7	<0.001
*Biochemical data*			
eGFR (mL/min/1.73 m^2^)	105.4 ± 28.8	89.1 ± 27.7	0.002 *
ALT (U/L, normal ≤36)	22.5 ± 10.8	24.2 ± 20	0.920
Total bilirubin (mg/dL, normal ≤1.3)	0.8 ± 0.4	0.4 ± 0.4	0.392
Uric acid (mg/dL, normal <7.0)	5.2 ± 1.7	4.9 ± 1.5	0.278
Glucose (AC) (mg/dL, normal: 70–100)	115.2 ±16.5	96.3 ± 41.2	0.042 *
** *Nutritional–inflammatory biomarkers* **			
Hb (g/dL)	12.0 ± 1.8	10.3 ± 1.5	<0.001 *
WBC (×10^3^/mm^3^)	6.6 ± 2.8	4.4 ± 3.0	0.001 *
Platelet count (×10^3^/mm^3^)	260.1 ± 91.4	212.2 ± 106.6	0.004 *
TLC (×10^3^/mm^3^)	1.8 ± 0.6	1.2 ± 0.6	<0.001 *
TNC (×10^3^/mm^3^)	4.0 ± 2.0	4.6 ± 2.3	<0.001 *
TMC (×10^3^/mm^3^)	0.5 ± 0.3	0.4 ± 0.3	0.352
Albumin (g/dL, normal: 3.5–5.5)	3.9 ± 0.4	3.8 ± 0.5	0.683
Prealbumin (g/dL, normal: 20–40)	25.3 ± 5.4	24.4 ± 8.5	0.467
Transferrin (g/dL normal: 200–360)	208.0 ± 34.6	198.8 ± 43.6	0.177
Total cholesterol (mg/dL, normal <200)	169.3 ± 41.8	167.8 ± 47.2	0.797
Triglyceride (mg/dL, normal <150)	160.5 ± 15.4	132.4 ± 59.8	<0.001 *
CRP (mg/L)	6.3 ± 7.7	13.8 ± 6.6	0.035 *
** *HLOP Metabolites* **			
Histidine (μM)	78.8 ± 16.2	75.5 ± 17.2	0.296
Leucine (μM)	128.7 ± 40.2	115.5 ± 25.3	0.005 *
Ornithine (μM)	122.5 ± 35.4	104.2 ± 31.8	0.001 *
Phenylalanine (μM)	63.1 ± 10.4	67.2 ± 19.5	0.357
** *DXA-derived measurement, kg* **			
LBM	43.8 ± 5.9	40.6 ± 5.7	<0.001 *
TFM	17.5 ± 7.3	16.7 ± 7.0	0.036 *
ASM	18.5 ± 3.3	17.0 ± 3.0	<0.001 *

* Comparing the differences between the start and end of CCRT for each variable; *p* < 0.05, statistically significant. The Wilcoxon signed-rank test was used for ∆Hb%, CRP, ∆ uric acid%, total bilirubin, mean time for feeding tube placement, dose, fraction and days of radiotherapy, cisplatin dose, all DXA-related measurements, all metabolites, and the number of grade 3/4 toxicities. Paired *t*-tests were used for other continuous variables. Abbreviations—NIBs: nutrition–inflammation biomarkers; DXA: dual-energy X-ray absorptiometry; SD: standard deviation; LAHNSCC: locally advanced head and neck squamous-cell carcinoma; CCRT: concurrent chemoradiotherapy; BW: body weight; BMI: body mass index; eGFR: estimated glomerular filtration rate; ALT: alanine aminotransferase; Hb: hemoglobin; WBC: white blood cell; TLC: total lymphocyte count; TNC: total neutrophil count; TMC: total monocyte count; CRP: C-reactive protein; LBM: lean body mass; TFM: total fat mass; ASM: appendicular skeletal mass; HLOP: histidine, leucine, ornithine, and phenylalanine.

**Table 3 cancers-14-03112-t003:** Clinical features of nine LAHNSCC patients with treatment interruptions during CCRT.

Case	Sex	Age	Tumor Site	TNM Stage	CCRT	Treatment Interruption Type *	RT (Dose, frx, Duration)	Cisplatin (Dose, % Completion)	Causes of Treatment Interruption
1	Male	59	Hypopharynx	T4aN2cM0, IVA	Primary	RT break and cisplatin break	72 Gy, 36 frx, 66 days	200 mg/m^2^, 83.3%	Grade 3/4 toxicities: infection and mucositis
2	Male	65	Hypopharynx	T2N2bM0, IVA	Primary	RT break and cisplatin break	72 Gy, 36 frx, 60 days	135 mg/m^2^, 56.3%	Grade 3/4 toxicities: infection, mucositis, and neutropenia
3	Male	50	Tongue	T2N2bM0, IVA	Adjuvant	RT break and cisplatin break	60 Gy, 30 frx, 48 days	200 mg/m^2^, 83.3%	Grade 3/4 toxicities: mucositis, emesis, and neutropenia
4	Male	58	Hypopharynx	T4bN2cM0, IVB	Primary	RT break	72 Gy, 36 frx, 62 days	240 mg/m^2^, 100%	Grade 3/4 toxicities: infection and neutropenia,
5	Female	64	Tonsil	T1N2bM0, IVA	Primary	RT break and cisplatin break	66 Gy, 33 frx, 53 days	200 mg/m^2^, 83.3%	Grade 3/4 toxicities: infection, mucositis, and anemia
6	Male	61	Retromolar	T4aN0M0, IVA	Adjuvant	RT break	60 Gy, 30 frx, 49 days	240 mg/m^2^, 100%	Grade 3/4 neutropenia
7	Male	29	Larynx	T1N2bM0, IVA	Primary	RT break	72 Gy, 36 frx, 56 days	240 mg/m^2^, 100%	Grade 3/4 thrombocytopenia
8	Male	47	Buccal mucosa	T3N1M0, III	Adjuvant	RT break	62 Gy, 31 frx, 51 days	240 mg/m^2^, 100%	Grade 3/4 mucositis
9	Male	44	Larynx	T3N3bM0, IVB	Primary	RT break	66 Gy, 33 frx, 56 days	240 mg/m^2^, 100%	Grade 3/4 toxicities: infection and neutropenia

Abbreviations: LAHNSCC, locally advanced head and neck squamous-cell carcinoma; CCRT, concurrent chemoradiotherapy; RT, radiotherapy; frx, fractions. * RT break was defined as an RT interruption of ≥5 days, and cisplatin break was defined as non-administration of planned weekly cisplatin (40 mg/m^2^).

**Table 4 cancers-14-03112-t004:** Univariate and multivariate logistic regression analyses of risk factors associated with treatment interruption of 44 patients with LAHNSCC completing CCRT.

Variables	Univariate	Multivariate	
	*p*-Value *	Odds Ratio (95% Confidence Interval)	*p*-Value *
*Clinicopathological factor*			
Age	0.534		
Sex (ref: female)	0.324		
Tumor location (ref: non-oral-cavity)	0.122		
TNM stage (ref: IV)	0.353		
T status (ref: T3-4)	0.142		
N status (ref: N2-3)	0.406		
Histological grade (ref: poorly differentiated)	0.108		
ECOG performance status (ref: 2)	0.115		
Smoking (ref: yes)	0.565		
Alcohol (ref: yes)	0.648		
Betel nut (ref: yes)	0.574		
Tracheostomy (ref: yes)	0.268		
PG-SGA before CCRT (ref: severe)	0.524		
*Biochemical data*			
Before CCRT			
eGFR (mL/min/1.73 m^2^)	0.926		
ALT (U/L)	0.214		
Total bilirubin (mg/dL)	0.146		
Uric acid (mg/dL)	0.248		
Sugar (AC) (mg/dL)	0.321		
Treatment-interval change (%)			
∆eGFR% ⁑	0.083		
∆ALT% ⁑	0.236		
∆Total bilirubin% ⁑	0.853		
∆Uric acid% ⁑	0.137		
∆Sugar (AC) ⁑	0.133		
*Anthropometric data and blood NIB data*			
Before CCRT			
BW (kg)	0.613		
BMI (kg/m^2^)	0.321		
Hb (g/dL)	0.765		
WBC (×10^3^ cells/mm^3^)	0.223		
Platelet count (×10^3^/mm^3^)	0.757		
TLC (×10^3^ cells/mm^3^)	0.455		
TNC (×10^3^/mm^3^)	0.234		
TMC (×10^3^/mm^3^)	0.265		
Albumin (g/dL)	0.796		
Prealbumin (g/dL)	0.516		
Transferrin (g/dL)	0.944		
Total cholesterol (mg/dL)	0.910		
Triglyceride (mg/dL)	0.648		
CRP (mg/dL)	0.133		
Treatment-interval change (%)			
∆BW% ⁑	0.786		
∆BMI% ⁑	0.351		
∆Hb% ⁑	0.190		
∆WBC% ⁑	0.934		
∆Platelet% ⁑	0.126		
∆TLC% ⁑	0.444		
∆TNC% ⁑	0.074		
∆TMC% ⁑	0.191		
∆Albumin% ⁑	0.031 *	0.906 (0.824–0.990)	0.038 *
∆Prealbumin% ⁑	0.306		
∆Transferrin% ⁑	0.109		
∆Cholesterol% ⁑	0.731		
∆Triglyceride% ⁑	0.776		
∆CRP% ⁑	0.410		
*DXA-related measurements*			
Before CCRT			
LBM (kg)	0.525		
TFM (kg)	0.753		
ASM (kg)	0.605		
Treatment-interval change (%)			
∆LBM% ⁑	0.412		
∆TFM% ⁑	0.545		
∆ASM% ⁑	0.034 *		
*HLOP Metabolites*			
Before CCRT			
Histidine (μM)	0.094		
Leucine (μM)	0.265		
Ornithine (μM)	0.130		
Phenylalanine (μM)	0.265		
Treatment-interval change (%)			
∆Histidine% ⁑	0.029 *	0.953 (0.911–0.980)	0.031 *
∆Leucine% ⁑	0.894		
∆Ornithine% ⁑	0.731		
∆Phenylalanine% ⁑	0.667		
*Mean daily calorie intake during CCRT (ref: ≥30)* †	0.591		
*Mean daily protein intake during CCRT ††*	0.868		
*Mean daily CHO intake during CCRT ††*	0.580		
*Mean daily fat intake during CCRT ††*	0.747		
*Feeding tube placement (no* vs. *yes)*	0.709		
*Mean days of feeding tube placement during CCRT*	0.741		
*CCRT factor*			
RT dose (Gy)	0.802		
RT fractions	0.390		
Cisplatin dose	0.027 *		
*CCRT-induced grade 3/4 toxicity*			
Dermatitis (ref: yes)	0.966		
Pharyngitis (ref: yes)	0.999		
Mucositis (ref: yes)	0.026 *		
Infection (ref: yes)	0.665		
Emesis (ref: yes)	0.574		
Anemia (ref: yes)	0.946		
Neutropenia (ref: yes)	0.834		
Thrombocytopenia (ref: yes)	0.968		
Number of grade 3/4 toxicity	0.016 *		

† Unit: Kcal/kg/day †† unit: g/kg/day; * indicates a significant *p*-value < 0.05. Abbreviations: CCRT: concurrent chemoradiotherapy; TNM: tumor node metastasis; ECOG: Eastern Collaboration Oncology Group; HN-CCI: head and neck Charlson Comorbidity Index; RT: radiotherapy; PG-SGA: patient-generated subjective global assessment; eGFR: estimated glomerular filtration rate; ALT: alanine transaminase; NIBs: nutritional–inflammatory biomarkers; BMI: body mass index; BW: body weight; Hb: hemoglobin; WBC: white blood cell count; TLC: total lymphocyte count; TNC: total neutrophil count; TMC: total monocyte count; CRP: C-reactive protein; DXA: dual-energy X-ray absorptiometry; LBM: lean body mass; TFM: total fat mass; ASM: appendicular skeletal mass; HLOP: histidine, leucine, ornithine, and phenylalanine. ⁑ ∆ indicates the value obtained by subtracting the pretreatment value from the post-treatment value; % indicates (∆ value/the pretreatment value) × 100%.

**Table 5 cancers-14-03112-t005:** Univariate and multivariate associations of risk factors with interval changes in albumin and histidine levels over the CCRT course in 44 patients with LAHNSCC completing CCRT.

Variables	∆Albumin%			∆Histidine%		
	Univariate	Multivariate		Univariate	Multivariate	
	*p*-Value *	Coefficient (95% CI)	*p*-Value *	*p*-Value *	Coefficient (95% CI)	*p*-Value *
*Clinicopathological factors*						
Age	0.145			0.391		
Sex (male vs. female)	0.484			0.263		
Tumor location (OC vs. NOC)	0.858			0.451		
TNM stage (III vs. IVA vs. IVB)	0.094			0.345		
T status (T1-2 vs. T3-4)	0.709			0.087		
N status (N0-1 vs. N2-3)	0.536			0.518		
Histological grade (1 vs. 2 vs. 3)	0.513			0.026 *		
Smoking (no vs. yes)	0.082			0.652		
Alcohol (no vs. yes)	0.869			0.240		
Betel nut (no vs. yes)	0.977			0.785		
ECOG performance status (0:1:2)	0.836			0.199		
Tracheostomy (no vs. yes)	0.069			0.199		
PG-SGA before CCRT (none: moderate: severe)	0.422			0.531		
*Biochemical data*						
Before CCRT						
eGFR (mL/min/1.73 m^2^)	0.896			0.101		
ALT (U/L)	0.934			0.377		
Total bilirubin (mg/dL)	0.068			0.581		
Uric acid (mg/dL)	0.049 *			0.141		
Sugar (AC) (mg/dL)	0.923			0.389		
Treatment-interval change (%)						
∆eGFR% ⁑	0.019 *			0.629		
∆ALT% ⁑	0.076			0.271		
∆Total bilirubin% ⁑	0.157			0.435		
∆Uric acid% ⁑	<0.001 *	0.085 (0.022~0.147)	0.009 *	0.068		
∆Sugar (AC) ⁑	0.194			0.510		
*Anthropometric data and blood NIBs data*						
Before CCRT						
BW (kg)	0.913			0.337		
BMI (kg/m^2^)	0.839			0.226		
Hb (g/dL)	0.060			0.084		
WBC (×10^3^ cells/mm^3^)	0.666			0.323		
Platelet count (×10^3^/mm^3^)	0.406			0.046 *		
TLC (×10^3^ cells/mm^3^)	0.289			0.953		
TNC (×10^3^/mm^3^)	0.874			0.288		
TMC (×10^3^/mm^3^)	0.831			0.298		
Albumin (g/dL)	0.031 *			0.877		
Prealbumin (g/dL)	0.690			0.237		
Transferrin (g/dL)	0.009 *			0.736		
Total cholesterol (mg/dL)	0.104			0.287		
Triglyceride (mg/dL)	0.432			0.307		
CRP (mg/dL)	0.342			0.003 *		
Treatment-interval change (%)						
∆BW% ⁑	0.812			0.421		
∆BMI% ⁑	0.464			0.113		
∆Hb% ⁑	<0.001 *			0.014 *		
∆WBC% ⁑	0.247			0.561		
∆Platelet% ⁑	0.087			0.426		
∆TLC% ⁑	0.260			0.827		
∆TNC% ⁑	0.178			0.428		
∆TMC% ⁑	0.726			0.230		
∆Albumin% ⁑	----			0.692		
∆Prealbumin% ⁑	0.003 *			0.229		
∆Transferrin% ⁑	<0.001 *	0.360 (0.237~0.483)	<0.001 *	0.458		
∆Cholesterol% ⁑	<0.001 *			0.365		
∆Triglyceride% ⁑	0.616			0.552		
∆CRP% ⁑	0.077			0.006 *	−0.004 (−0.007~−0.0.001)	0.019 *
*DXA-related measurements*						
Before CCRT						
LBM (kg)	0.726			0.605		
TFM (kg)	0.570			0.295		
ASM (kg)	0.938			0.944		
Treatment-interval change (%)						
∆LBM% ⁑	0.973			0.065		
∆TFM% ⁑	0.176			0.590		
∆ASM% ⁑	0.042 *			0.035 *		
*HLOP Metabolites*						
Before CCRT						
Histidine (μM)	0.056			<0.001 *		
Leucine (μM)	0.531			0.295		
Ornithine (μM)	0.087			0.307		
Phenylalanine (μM)	0.930			0.115		
Treatment-interval change (%)						
∆Histidine% ⁑	0.692			----		
∆Leucine% ⁑	0.305			0.015 *		
∆Ornithine% ⁑	0.344			0.022 *		
∆Phenylalanine% ⁑	0.442			0.001 *	0.314 (0.065~0.564)	0.015 *
*Mean daily calorie intake during CCRT †*	0.429			0.577		
*Mean daily protein intake during CCRT ††*	0.406			0.734		
*Mean daily CHO intake during CCRT ††*	0.429			0.520		
*Mean daily fat intake during CCRT ††*	0.428			0.582		
*Feeding tube placement (no* vs. *yes)*	0.245			0.583		
*Mean days of feeding tube placement during CCRT*	0.609			0.629		
*CCRT factor*						
RT dose (Gy)	0.247			0.746		
RT fractions	0.391			0.900		
RT duration (days)	0.474			0.201		
Cisplatin dose (mg/m^2^)	0.323			0.677		
*CCRT-induced grade 3/4 toxicity*						
Non-hematological						
Dermatitis (no vs. yes)	0.695			0.404		
Pharyngitis (no vs. yes)	0.908			0.237		
Mucositis (no vs. yes)	0.325			0.282		
Infection (no vs. yes)	0.482			0.247		
Emesis (no vs. yes)	0.563			0.495		
Hematologic						
Anemia (no vs. yes)	0.509			0.563		
Neutropenia (no vs. yes)	0.151			0.734		
Thrombocytopenia (no vs. yes)	0.532			0.858		
Number of grade 3/4 toxicity	0.016 *					

† Unit: Kcal/kg/day †† unit: g/kg/day; * Indicates a significant *p*-value < 0.05. Univariate analysis: simple linear regression model for all continuous variables; Mann–Whitney test for sex, T status, N status, smoking, alcohol consumption, betel nut usage, tracheostomy, and feeding tube placement; Kruskal–Wallis test for TNM stage, histological grade, ECOG performance status, and PG-SGA. ⁑ ∆ indicates the value obtained by subtracting the pretreatment value from the post-treatment value; % indicates (∆ value/the pretreatment value) ×100%. Abbreviations: NIBs: nutritional or inflammatory biomarkers; LAHNSCC: locally advanced head and neck squamous-cell carcinoma; CCRT: concurrent chemoradiotherapy; CI: confidence interval; OC: oral cavity; NOC: non-oral-cavity; TNM: tumor node metastasis; ECOG: Eastern Collaboration Oncology Group; RT: radiotherapy; PG-SGA: patient-generated subjective global assessment; NIBs: nutritional–inflammatory biomarkers; BMI: body mass index; BWL: body weight loss; Hb: hemoglobin; WBC: white blood cell count; TLC: total lymphocyte count; TNC: total neutrophil count; TMC: total monocyte count; CRP: C-reactive protein; DXA: dual-energy X-ray absorptiometry; LBM: lean body mass; TFM: total fat mass; ASM: appendicular skeletal mass; HLOP: histidine, leucine, ornithine, and phenylalanine.

## Data Availability

The data presented in this study are available upon request from the corresponding author.

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
