# Peer review of "Treatment-Interval Changes in Serum Levels of Albumin and Histidine Correlated with Treatment Interruption in Patients with Locally Advanced Head and Neck Squamous Cell Carcinoma Completing Chemoradiotherapy under Recommended Calorie and Protein Provision"

_cancers, 2022, doi:10.3390/cancers14133112_

Round 1

Reviewer 1 Report

This study investigated the impact of the treatment-interval changes in serum levels albumin and histidine on locally advanced head and neck squamous cell carcinoma.  The aim of the study is interesting, but it seems complicated to draw clear conclusion on this very small number of patients. 

Reviewer 2 Report

This present article by Wang et al helps to understand the calorie and protein intake risk factors for treatment interruption in patients with locally advanced head and neck squamous cell carcinoma (LAHNSCC) following concurrent chemoradiotherapy (CCRT). They shed light on changes in patients' serum albumin and histidine levels during treatment intervals (TI). These observations would further help to understand the requirement of nutrients during TI. Overall the manuscript is concise and clear. I am in principle supportive of accepting this work for publication. 

Author Response

We sincerely thank the reviewer for the interest in our patient cohort and results.

Reviewer 3 Report

The reviewed report has come from a laborious clinical study concerning a potential failure of head and neck cancer. The authors mentioned two reasons of interruptions of concurrent chemoradiotherapy. First is connected with untolerable toxicity, the second of instrumental, other technical, organization of supply unplanned difficulties. This is a clinical reality. The problem is already known and it was studied on many ways. The authors studied a huge number of clinical, anthropometric and biochemical variables that is a primary advantage of the study. The study has been approved by ethical requirements. It required also a discussion related to many points that involved a long list of citations.

As the manuscript answers questions of clinicians and is well written as suggest to accept it as it is.

Author Response

We deeply appreciate the reviewer’s comment and support to the idea and results of this manuscript.